# Deprescribing Central Nervous System-Active Medications Among Community-Dwelling Older Adults with Dementia in Primary Care: A Feasibility Study

**DOI:** 10.3390/ijerph22111611

**Published:** 2025-10-22

**Authors:** Elizabeth A. Phelan, Benjamin H. Balderson, Monica M. Fujii, Vina F. Graham, Mary Kay Theis, Shelly L. Gray

**Affiliations:** 1Department of Medicine, Division of Gerontology and Geriatric Medicine, School of Medicine, University of Washington, Seattle, WA 98104, USA; 2Department of Health Systems and Population Health, School of Public Health, University of Washington, Seattle, WA 98195, USA; 3Kaiser Permanente Washington Health Research Institute, Kaiser Permanente Washington, Seattle, WA 98101, USA; benjamin.h.balderson@kp.org (B.H.B.); vina.f.graham@kp.org (V.F.G.); kaytheis@hotmail.com (M.K.T.); 4Department of Pharmacy, School of Pharmacy, University of Washington, Seattle, WA 98195, USA; slgray@uw.edu

**Keywords:** aged, dementia, falls, potentially inappropriate medication list, deprescriptions, primary health care

## Abstract

Central nervous system (CNS)-active medications pose serious health risks for older adults with dementia but are nonetheless commonly used. Few deprescribing interventions have focused on people with dementia. We conducted a one-arm pilot study in six primary care practices of an integrated healthcare system between February and August 2023. The deprescribing intervention consisted of patient/care partner education and self-management materials and provider decision support. Participants were aged 60+ with diagnosed dementia and prescribed at least one CNS-active medication for three or more months of the six-month period prior to study start. We assessed feasibility and acceptability of the intervention and feasibility of ascertaining medication discontinuation and medically treated falls. The intervention was delivered to all (*N* = 114) eligible participants; their mean age was 80 ± 9 years; 72% were female and 13% non-White. Intervention acceptability, assessed by Weiner’s Acceptability of Intervention measure, was rated 3.5/5 (range 1–5; higher scores indicate higher acceptability). Among baseline antipsychotic users (*N* = 89), 39 (43.8%) had discontinued at follow-up. Among baseline tricyclic antidepressant users (*N* = 11), 6 (54.5%) had discontinued at follow-up. Among baseline skeletal muscle relaxant users (*N* = 3), 2 (66.7%) had discontinued at follow-up. Among baseline benzodiazepine users (*N* = 3), 1 (33.3%) had discontinued at follow-up. Among baseline opioid users (*N* = 13), 1 (7.7%) had discontinued at follow-up. Medically treated falls occurred among 22% at baseline vs. 21% at follow-up. The intervention is feasible and acceptable and may achieve meaningful reduction in CNS-active medication prescriptions. Findings support a controlled trial with sufficient power to assess effects on relevant clinical outcomes.

## 1. Introduction

Falls among older adults are a major public health concern given their multiple adverse consequences, including severe injury, functional decline, nursing home placement, and mortality [1]. Older people with dementia (OPWD) have eight to ten times more incident falls compared to age-matched peers without dementia [2,3]. OPWD are also less likely than those without dementia to make a full functional recovery after a fall-related injury [4], and falls increase burden on care partners [5,6].

Medications and particularly those that affect the central nervous system (CNS) are a key modifiable risk factor for falls [7]. Research has found that reducing CNS-active medications can reduce falls [8]. CNS-active medications are considered potentially inappropriate for older adults, especially for OPWD who have heightened sensitivity to the effects of CNS-active medications [9], and guidelines recommend avoiding their use [10]. However, use remains common and is increased among OPWD [11,12].

Deprescribing is the process of withdrawing inappropriate medication, supervised by a health care professional, with the goal of managing polypharmacy and improving outcomes [13]. Dose reduction, switching to a safer alternative, or implementing non-pharmacologic strategies are important outcomes of deprescribing [14]. Limited published data suggest that deprescribing may result in reduced out-of-pocket drug costs, improved quality of life, and reduced risk of adverse events and mortality [15]. Few deprescribing interventions have targeted community-dwelling OPWD [16,17], and few have examined the effect of deprescribing CNS-active medications on clinical outcomes, in particular fall-related medical care [18].

To address this guideline–practice gap, we conducted a pilot study to assess the feasibility of extending a deprescribing intervention to reduce use of CNS-active medications, previously tested with cognitively intact older adults [19], to OPWD and their care partners. The intervention, referred to as STOP-FALLS, consists of patient/care partner education, self-management guidance, and provider decision support. Intervention materials describe the risks of CNS-active medications and safer alternatives, in particular, non-pharmacologic strategies to address the symptoms for which these medications are often prescribed. The aims of the pilot study were to assess (1) feasibility of reaching OPWD and their care partners, (2) acceptability of the intervention to OPWD and their care partners, (3) feasibility of ascertaining discontinuation of CNS-active medications targeted by the intervention and medically treated falls.

## 2. Materials and Methods

**Study Setting.** The study was conducted at Kaiser Permanente Washington (KPWA), an integrated delivery system in the Northwest United States. KPWA is one of eight regional Kaiser health plans nationally. KPWA provides comprehensive medical care and coverage to ~700,000 individuals across Washington State, including ~100,000 Medicare beneficiaries. There are 25 primary clinics across the state, and all clinics use the electronic health record known as Epic.

**Participants.** Participants were identified from KPWA’s automated administrative and pharmacy databases. Inclusion criteria were aged ≥60 years, diagnosed dementia, based on either a dementia diagnosis code [20] or prescription for a cholinesterase inhibitor or memantine, receiving primary care at a KPWA integrated group practice outpatient clinic, and taking at least one CNS-active medication on a chronic (≥3 month) basis, as determined by pharmacy fill records. Exclusion criteria were residence in a skilled nursing facility, cancer diagnosis, or on hospice or palliative care.

**Intervention Content.** The intervention consisted of theory-driven, educational brochures about the CNS-active medications targeted by the intervention, handouts with self-management strategies for symptom management, and provider decision support in the form of an evidence-based pharmaceutical opinion (EBPO) [21]. Six classes of medications that contribute to falls in older adults were the target of the intervention: antipsychotics, first-generation antihistamines, sedative-hypnotics (benzodiazepines and Z-drugs), opioids, skeletal muscle relaxants, and tricyclic antidepressants.

Prior to the pilot trial, patient-facing intervention materials were vetted with a convenience sample of dementia care partners (*N* = 7) of OPWD receiving healthcare through KPWA through a series of one-on-one, semi-structured interviews conducted by study investigators. Care partner input guided minor revisions to the materials.

Provider materials referenced the latest Beers criteria and other literature detailing the risks associated with use of the particular drug class by older adults. They also included KPWA resources and referenced KPW tapering guidance, where available. All decision support materials were vetted by a group of KPWA District Medical Directors and revised based on that input.

**Intervention Implementation.** Eligible participants were mailed a letter describing the study, an educational brochure about the medication that they were prescribed, and a handout on non-pharmacologic approaches to self-management of symptoms for which that medication is typically prescribed. The cover letter encouraged participants to share the information with anyone who assisted them with their medical care and medications. The date of mailing was defined as the index date for evaluation purposes.

Synchronous with the mailing, decision support was sent to the participant’s primary care provider (PCP) via the staff message feature in the electronic health record. The message identified the patient and the medication that they had been prescribed. The message contained a Uniform Resource Locator (URL) to the study website where decision support pertaining to that target medication class could be found.

Given the brief duration of the study, a single mailing that included the study research questionnaire, a medication brochure for one of the target medication classes, and an antihistamine brochure was conducted. The rationale for mailing an antihistamine brochure to all participants was that most antihistamines are able to be obtained without a prescription, and their use among KPWA patients has been found to be high [22].

**Evaluation.** The evaluation was conducted between March and December 2023 using pharmacy fill records, electronic healthcare utilization files of the health plan, and the research questionnaire mailed to participants. A pre–post design and descriptive statistics were used. Feasibility was assessed by successful mailed delivery of the intervention materials, operationalized as the percent of mailings returned to sender due to being undeliverable. Acceptability was measured by the 4-item Acceptability of Intervention (AIM) instrument [23], ascertained via the mailed questionnaire. Response options for each item on the AIM range from 1–5 (1 = completely disagree, 2 = disagree, 3 = neither agree nor disagree, 4 = agree, 5 = completely agree); higher scores indicate higher acceptability. Prescription fill data were obtained from the KPWA electronic pharmacy database. Baseline medication use was defined as a prescription fill of a target medication for a minimum of 70 of the 90 days prior to the index date. Medication discontinuation was defined as no evidence of a pharmacy fill at month 6 of follow-up. Patients enrolled in the integrated group practice clinics of the KPWA obtain all prescription medications at a KPWA pharmacy, making absence of a fill a reasonable proxy for discontinuation. Episodes of medical care for falls were extracted from the KPWA virtual data warehouse for 6 months prior and 6 months post index using methods described previously [19]. The study protocol was reviewed by the ADVARRA Institutional Review Board and approval obtained.

**Sample Size.** Using the framework for sample size calculation for pragmatic pilot trials [24] and the primary endpoint of feasibility, assuming the upper boundary of the “red” zone was 50% and the lower boundary of the “green” zone was 75% (designating unacceptable and acceptable feasibility respectively), the sample size required for analysis given 95% power and one-sided 5% alpha was determined to be *N* = 42 (intervention arm only). We increased the sample size to ensure sufficient (>90%) power for other outcomes.

**Data Analysis.** Baseline characteristics of the study sample and acceptability data were analyzed using descriptive statistics. Continuous data are reported as mean ± standard deviation; categorical data are presented as number (percent) of the study sample. Because the study focused on feasibility, inferential statistical testing was not performed; only descriptive statistics are presented.

## 3. Results

The intervention was delivered to all (*N* = 114) eligible participants. The mean age was 80 ± 9 years; 72% were female; 13% were non-White, and 30% were frail. Over half had hypertension, and nearly half were diagnosed with depression. Most were prescribed just one of the CNS-active medication classes examined in the pilot study. Prescriptions for other classes of CNS-active medications not targeted by the intervention, in particular antidepressants and gabapentinoids, were prevalent. See Table 1 for additional information on the study sample.

Fifteen participants (13%) returned the mailed questionnaire. Intervention acceptability was rated as 3.5/5, on average (range 1–5; standard deviation 0.95; higher scores indicate higher acceptability). Notable comments from respondents included, “we are taking the letter to the next appointment to talk to the psychiatrist about it” (*n* = 1) and, from another (*n* = 1), “why give this to an elderly woman with dementia? Why was she given this?”

Analyses of pharmacy fill data (Table 2) found that at baseline, among the target medication classes, antipsychotic prescriptions were the most frequent (prescribed to 78% of participants), followed by opioids (11%) and tricyclic antidepressants (10%). Just 3% were prescribed a benzodiazepine or a muscle relaxant. None were prescribed a first-generation antihistamine or a Z-drug (zolpidem, eszopiclone, zaleplon).

Among baseline antipsychotic users (*N* = 89), 39 (43.8%) had discontinued at follow-up. Among baseline tricyclic antidepressant users (*N* = 11), 6 (54.5%) had discontinued at follow-up. Among baseline skeletal muscle relaxant users (*N* = 3), 2 (66.7%) had discontinued at follow-up. Among baseline benzodiazepine users (*N* = 3), 1 (33.3%) had discontinued at follow-up. Among baseline opioid users (*N* = 13), 1 (7.7%) had discontinued at follow-up.

Medically treated falls occurred among 22% at baseline vs. 21% at follow-up.

## 4. Discussion

This study suggests that a deprescribing intervention targeting older adults with dementia is feasible and acceptable and may result in a reduction in prescription of CNS-active medications. We observed a reduction in prescriptions for all medication classes examined. No change in medically treated falls was observed, which was not unexpected, given the short duration of follow-up and the small sample size.

Effective strategies to reduce prescription of CNS-active medications for older adults with dementia are a pressing healthcare priority, because use of these medications is exceedingly common, and use confers an increased likelihood of multiple adverse outcomes, including stroke, falls and fractures, hospitalization, and death [25]. In the United States, where the present study was conducted, nearly three-quarters (73.5%) of older adults with dementia residing in the community may be prescribed a high-risk medication [11]. Recent large deprescribing trials focusing on OPWD residing in the community have been ineffective in changing prescription of high-risk medications [17].

Most studies assessing the impact of deprescribing interventions targeting OPWD have been conducted in long-term care facilities, and few have examined the intervention’s effect on clinical outcomes (e.g., falls) [26]. Our pilot trial is one of the first deprescribing studies to address a clinically relevant health outcome among OPWD residing in the community.

Few deprescribing trials have engaged the person with dementia and/or their care partner [16], although evidence suggests that care partners would like to be involved in medication management [27]. Managing one’s own medications is one of the first functional abilities that becomes impaired in people with dementia, and medication oversight then often falls to the care partner. Care partners have an important role in advocating for decisions regarding medications that are concordant with what matters to the older person with dementia. Our intervention was guided by the Chronic Care Model [28], in which an informed, empowered patient has productive interactions with a prepared, proactive practice team to improve health outcomes. The patient-facing materials were designed to engage and activate patients and care partners to participate in deprescribing discussions, and the clinician-facing components were designed to facilitate those discussions and medication tapering, with pharmacist guidance on tapering upon PCP request. Our study provides a signal that the involvement of people with dementia and their care partners may help drive medication discontinuation. Controlled studies are needed to confirm these results.

Care partner information (e.g., name, contact information, relationship to the person with dementia) was not available in the health system at the time the present study was conducted. While we were able to deliver our intervention to those with dementia without identifying their care partner(s), our qualitative findings suggest that for some care partners, an intervention delivered to the person with dementia may raise concerns. Methods to identify care partners pragmatically from health system records are an area of active investigation [29]. Care partner identification is an important consideration for future health-system-embedded pragmatic trials involving people with dementia.

Strengths of the present study include design features, including its pragmatic approach to implementation of the intervention and involvement of the person with dementia and their care partner, and its data collection capacity, namely complete capture of prescription fills and healthcare utilization. The latter was possible due to the integrated delivery system where the study was conducted, with its electronic data on prescription medication dispensing and healthcare utilization.

Study limitations include the absence of a control group, no pragmatic means by which to identify dementia care partners or assess whether clinicians accessed or engaged with decision support, the predominantly White study sample, and the low response rate to the mailed research questionnaire. The mailed questionnaire was anonymous, which precluded re-mailing to non-respondents to attempt to increase the response rate. As this was a feasibility study with a single-group, pre–post design, inferences cannot be made regarding the significance of changes in clinical outcomes. Another limitation is that dose reduction of target medications was not assessed as part of the pilot study. Dose reduction of high-risk medications is an important strategy to reduce the risk of falls, and a large-scale deprescribing trial should examine dose reduction as an outcome measure. These limitations notwithstanding, it was nonetheless feasible to complete the pilot without identifying care partners, obtain perceptions of the acceptability of the intervention, and accurately quantify prescription fills and medically treated falls. The qualitative responses from those who did return the research questionnaire will be useful for guiding future implementation efforts.

Future research needs to involve a randomized study design with a control group and a sufficient sample size to assess clinically relevant outcomes. In the context of a large-scale trial, process measures to elucidate clinician engagement with intervention materials could be collected and a range of medication outcomes (e.g., dose reduction, sustained discontinuation) could be assessed. Analyses of fall rates stratified by number and type of high-risk medication would also be informative.

## 5. Conclusions

Health-system-embedded deprescribing strategies targeting older people with dementia and their care partners are feasible and acceptable and may reduce prescription of CNS-active medications. Findings lend support for a controlled trial with sufficient power to assess effects on relevant clinical outcomes.

## Figures and Tables

**Table 1 ijerph-22-01611-t001:** Demographic and health characteristics of study participants (*N* = 114).

Characteristic	Mean ± SD or *N* (%)
Age, years	79.9 (9)
Age group, years	
60–64	7 (6)
65–74	25 (22)
75–84	49 (43)
85+	33 (29)
Female	82 (72)
Race ^a^	
Native Hawaiian/Pacific Islander	0 (0)
American Indian/Alaska Native	1 (1)
Asian	6 (5)
Black or African American	2 (2)
White	99 (87)
Other race ^b^	2 (2)
Multiple races	2 (2)
Race unknown or not reported	2 (2)
Hispanic ethnicity	4 (4)
Chronic condition	
Alcohol use disorder	4 (4)
Anxiety	36 (32)
Bipolar affective disorder	7 (6)
Congestive heart failure	11 (10)
Depression	51 (45)
Diabetes	24 (21)
Hearing loss	16 (14)
Hypertension	64 (56)
Insomnia	29 (25)
Musculoskeletal pain	38 (33)
Orthostatic hypotension	1 (1)
Osteoarthritis	16 (14)
Osteoporosis	15 (13)
Parkinson’s disease	5 (4)
Peripheral neuropathy	8 (7)
Schizophrenia	2 (2)
Frailty state ^a,c^	
Non-frail	12 (11)
Pre-frail	68 (60)
Frail	34 (30)
Number of classes of target medications	
1	109 (96)
2	5 (4)
Other classes of CNS-active medications	
Antidepressant (non tricyclic) ^d^	93 (82)
Gabapentinoid	21 (18)
Other sedative-hypnotic ^e^	1 (2)

Abbreviations: SD = Standard deviation. ^a^ Percentages may not total 100 due to rounding. ^b^ Other race as reported by the participant in their electronic medical record field (e.g., Middle Eastern; North African). ^c^ Frailty assessed using Kim’s Claims-Based Frailty Index. ^d^ Non tricyclic antidepressants included citalopram, escitalopram, fluoxetine, fluvoxamine, paroxetine, sertraline, vilazodone, bupropion, desvenlafaxine, duloxetine, levomilnacipran, milnacipran, venlafaxine, nefazodone, trazodone, vortioxetine, and mirtazapine. ^e^ Other sedative-hypnotics include chloral hydrate, meprobamate, ramelteon, trazodone, and low-dose (3 mg, 6 mg) doxepin.

**Table 2 ijerph-22-01611-t002:** Prevalence of CNS-active medication class use among the study sample.

	Baseline ^a^	Follow-Up ^a^
Medication Class	Number (%) with Prescription	Number (%) with Prescription
Antipsychotic	89 (78)	51 (45) ^b^
Antihistamine	0 (0)	0 (0)
Benzodiazepine	3 (3)	3 (3) ^c^
Z-drug	0 (0)	0 (0)
Opioid	13 (11)	12 (11)
Skeletal muscle relaxant	3 (3)	1 (1)
Tricyclic antidepressant	11 (10)	6 (5)

^a^ Calculated based on number with a prescription for the given class of medication/study sample N. ^b^ Of 89 using at baseline, 39 (43.8%) had discontinued; one participant not using at baseline filled a new prescription at 6 months. ^c^ Of 3 using at baseline, 1 had discontinued; one participant not using at baseline filled a new prescription at 6 months.

## Data Availability

The data presented in this article are not publicly available, because study participants have not directly consented to having their data used for research purposes, and because health plan data are proprietary. Requests to access the data should be directed to the Kaiser Permanente Washington Health Research Institute.

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
