# Peer review of "Deprescribing Central Nervous System-Active Medications Among Community-Dwelling Older Adults with Dementia in Primary Care: A Feasibility Study"

_ijerph, 2025, doi:10.3390/ijerph22111611_

Round 1

Reviewer 1 Report

Comments and Suggestions for Authors

The article addresses a topic of high clinical and scientific relevance: the deprescribing of drugs that act on the central nervous system, with relevance in populations with dementia. The approach adopted is clear, objective, and methodologically well-designed. The introduction adequately contextualizes the problem, providing the theoretical basis necessary to understand the importance of the study. The methodology is described rigorously and allows for the reproducibility of the work. The results are presented in a structured, clear manner consistent with the proposed objectives. The discussion is well supported by the current literature, highlighting in a balanced way the implications, limitations and potential of the study. As a feasibility study, the work satisfactorily meets the methodological and scientific criteria, providing a solid basis for the development of future, more comprehensive and robust research in this area.

The authors acknowledge the weaknesses of their work and present them in the text. However, some methodological aspects and the presentation of results could be clarified, namely:

- Better justify the criteria used for discontinuing therapy and how the absence of dispensing in pharmacy records ensures that the patient has actually discontinued therapy. Can the patient not purchase the medication at another pharmacy without this record?

- The results obtained in the pre- and post-intervention comparisons did not provide sufficient statistical evidence to confirm or refute the significance of the observed effects.

- The reduction in falls (22% vs. 21%) was marginal and should be discussed further, considering that it was one of the objectives of the study.

Author Response

- Better justify the criteria used for discontinuing therapy and how the absence of dispensing in pharmacy records ensures that the patient has actually discontinued therapy. Can the patient not purchase the medication at another pharmacy without this record?

RESPONSE: We have added the following to address this comment (line 129): “Patients enrolled in the integrated group practice clinics of KPWA obtain all prescription medications at a KPWA pharmacy, making absence of a fill a reasonable proxy for discontinuation.”

- The results obtained in the pre- and post-intervention comparisons did not provide sufficient statistical evidence to confirm or refute the significance of the observed effects.

RESPONSE: We have added the following to highlight this point (line 239): “As this was a feasibility study with a single-group, pre-post design, inferences cannot be made regarding the significance of changes in clinical outcomes.”

- The reduction in falls (22% vs. 21%) was marginal and should be discussed further, considering that it was one of the objectives of the study.

RESPONSE: We have added the following text (line 189): “No change in medically treated falls was observed, which was not unexpected, given the short duration of follow-up and the small sample size.”

Reviewer 2 Report

Comments and Suggestions for Authors

This paper presents feasibility results from an intervention to reduce CNS-active medications in older adults with dementia. This is an important and timely topic, and the manuscript provides sound evidence of early-stage intervention work that lays important groundwork for future research. My comments are below.

Sample clarity
In the results section, the manuscript states that the intervention was delivered to all eligible participants, who I assume represent the 15% of respondents mentioned in the limitations. First, it would strengthen the manuscript to report this response rate in the methods or results, rather than only in the limitations. In addition, because the intervention included both a patient component (mailed materials) and a provider component (a URL to the study website), it is unclear to what extent participants received both elements. While it seems safe to assume that respondents saw the mailed materials, there is no evidence presented that providers accessed or engaged with the website content. Questions such as whether providers clicked the URL, read the information, or incorporated it into practice may not be fully answerable here. Still, it would be useful to acknowledge this limitation explicitly and suggest it as an area for future study.

Statistical testing
The manuscript notes that “statistical testing was not performed.” While this phrase typically connotes that no inferential testing was conducted, the inclusion of descriptive statistics makes the statement potentially confusing. To improve clarity, I suggest rephrasing as: “Because the study focused on feasibility, inferential statistical testing was not performed; only descriptive statistics are presented.” As currently written, the phrasing aligns with feasibility study conventions, but this small change would reduce possible distraction for readers.

Acceptability ratings
I recommend including a dispersion parameter for the intervention acceptability rating; a standard deviation seems appropriate here (I appreciate that the range is already provided). Were text labels associated with the numerical scale? If so, reporting them would help contextualize the mean rating (e.g., a score of 3.5 could fall between “average acceptability” and “moderate acceptability”). If not, a brief discussion of how to interpret the numeric value would be helpful, as the score sits above the midpoint and indicates generally favorable attitudes.

Polypharmacy coding
I was somewhat surprised by the low rates of polypharmacy for the selected classes, although this is an encouraging finding. Because these medication classes contain multiple drugs, it is possible (though unlikely) that individuals had multiple prescriptions within the same class. It would be useful to clarify how this was handled in the coding; was only the first occurrence counted, or were all prescriptions within a class included? It is currently unclear if the number presented is the number of prescriptions for antipsychotics in the patient population or the number of patients with an antipsychotic prescription, they are not necessarily the same.

Dosing and Beers criteria
I would also encourage exploration of medication dosing. The Beers criteria provide recommendations for older adults, and describing the extent to which prescribing was consistent with these guidelines could be informative. Even if explored only for antipsychotics (given the sparseness of other classes), this could serve as another outcome measure. Deprescribing is important, but dose reduction may also represent a meaningful intervention effect.

Discussion language
Please ensure the discussion does not overstate the findings. The manuscript states: “…and may result in a reduction in prescription of CNS-active medications. We observed a reduction in prescriptions for all medication classes examined, including classes that are particularly challenging to deprescribe (opioids and benzodiazepines).” The first sentence is appropriately cautious. However, I recommend removing the clause “including classes that are particularly challenging to deprescribe.” Although descriptively true, the evidence presented is not sufficient to support this claim. In both explicitly mentioned categories, only one individual was deprescribed, and in one case a new prescription was initiated. Without inferential testing or stronger clinical evidence, this statement risks overstating the impact.

Additional analyses
If possible, examining fall rates by medication type pre/post (perhaps as a supplemental table) could add value. While the distributions may be uneven (given that most prescriptions were antipsychotics), this information could still provide context.

Future research
The future research section could be expanded. The limitations already noted suggest multiple avenues for subsequent investigation, and offering concrete examples would strengthen this section.

Engagement of persons with dementia and care partners
I appreciated the discussion around engaging both the person with dementia and their care partner, and encourage expansion of this point. The qualitative interviews provide compelling evidence; for example, one caregiver’s concern about sending extensive information directly to someone with dementia highlights the importance of involving care partners. In the conclusion, the manuscript states: “Care partner identification may not be necessary…however, engaging care partners may results in higher rates of deprescribing.” I recommend avoiding language that suggests care partners may be unnecessary. At least one caregiver explicitly questioned this approach, underscoring the value of incorporating care partners into the intervention.

Author Response

Sample clarity
In the results section, the manuscript states that the intervention was delivered to all eligible participants, who I assume represent the 15% of respondents mentioned in the limitations. First, it would strengthen the manuscript to report this response rate in the methods or results, rather than only in the limitations.

RESPONSE: The sample for the study was 114, as stated in the first sentence of the Results. We have added the response rate for the mailed questionnaire to the Results as recommended (line 163): “Fifteen participants (13%) returned the mailed questionnaire” and omitted it from the paragraph on limitations.

In addition, because the intervention included both a patient component (mailed materials) and a provider component (a URL to the study website), it is unclear to what extent participants received both elements. While it seems safe to assume that respondents saw the mailed materials, there is no evidence presented that providers accessed or engaged with the website content. Questions such as whether providers clicked the URL, read the information, or incorporated it into practice may not be fully answerable here. Still, it would be useful to acknowledge this limitation explicitly and suggest it as an area for future study.

RESPONSE: We have added the following text to address this suggestion (line 235): “no pragmatic means by which to… assess whether clinicians accessed or engaged with decision support...” We now also suggest it as an area for future research (line 253).

Statistical testing
The manuscript notes that “statistical testing was not performed.” While this phrase typically connotes that no inferential testing was conducted, the inclusion of descriptive statistics makes the statement potentially confusing. To improve clarity, I suggest rephrasing as: “Because the study focused on feasibility, inferential statistical testing was not performed; only descriptive statistics are presented.” As currently written, the phrasing aligns with feasibility study conventions, but this small change would reduce possible distraction for readers.

RESPONSE: We have made this edit as suggested (line 144).

Acceptability ratings
I recommend including a dispersion parameter for the intervention acceptability rating; a standard deviation seems appropriate here (I appreciate that the range is already provided). Were text labels associated with the numerical scale? If so, reporting them would help contextualize the mean rating (e.g., a score of 3.5 could fall between “average acceptability” and “moderate acceptability”). If not, a brief discussion of how to interpret the numeric value would be helpful, as the score sits above the midpoint and indicates generally favorable attitudes.

RESPONSE: We have added a standard deviation for the intervention acceptability rating as recommended (line 164). Text labels are associated with the numerical scale, and we now report those (line 124): “1=completely disagree, 2=disagree, 3=neither agree nor disagree, 4=agree, 5=completely agree.”

Polypharmacy coding
I was somewhat surprised by the low rates of polypharmacy for the selected classes, although this is an encouraging finding. Because these medication classes contain multiple drugs, it is possible (though unlikely) that individuals had multiple prescriptions within the same class. It would be useful to clarify how this was handled in the coding; was only the first occurrence counted, or were all prescriptions within a class included? It is currently unclear if the number presented is the number of prescriptions for antipsychotics in the patient population or the number of patients with an antipsychotic prescription, they are not necessarily the same.

RESPONSE: We assume that this comment refers to Table 1. As the table’s title describes, data in Table 1 are reported at the level of the participant. In this study, we focused on number of classes, not number of prescriptions within a given class. We have edited the table and other spots in the manuscript to clarify that the focus was on medication class. For example, the row label for the data on target medications now reads, “number of classes of target medications,” and the row label for the data presented just below now reads, “other classes of CNS-active medications.” Similarly, the title of Table 2 has been edited to “Prevalence of CNS-active medication class use among the study sample.”

Dosing and Beers criteria
I would also encourage exploration of medication dosing. The Beers criteria provide recommendations for older adults, and describing the extent to which prescribing was consistent with these guidelines could be informative. Even if explored only for antipsychotics (given the sparseness of other classes), this could serve as another outcome measure. Deprescribing is important, but dose reduction may also represent a meaningful intervention effect.

RESPONSE: We have added the following to the Discussion to address this point (line 240): “Another limitation is that dose reduction of target medications was not assessed as part of the pilot study. Dose reduction of high-risk medications is an important strategy to reduce the risk of falls, and a large-scale deprescribing trial should examine dose reduction as an outcome measure.”

Discussion language
Please ensure the discussion does not overstate the findings. The manuscript states: “…and may result in a reduction in prescription of CNS-active medications. We observed a reduction in prescriptions for all medication classes examined, including classes that are particularly challenging to deprescribe (opioids and benzodiazepines).” The first sentence is appropriately cautious. However, I recommend removing the clause “including classes that are particularly challenging to deprescribe.” Although descriptively true, the evidence presented is not sufficient to support this claim. In both explicitly mentioned categories, only one individual was deprescribed, and in one case a new prescription was initiated. Without inferential testing or stronger clinical evidence, this statement risks overstating the impact.

RESPONSE: We have omitted the clause as recommended (line 189).

Additional analyses
If possible, examining fall rates by medication type pre/post (perhaps as a supplemental table) could add value. While the distributions may be uneven (given that most prescriptions were antipsychotics), this information could still provide context.

RESPONSE: We appreciate this suggestion. This was a feasibility study, and it was not powered to examine differences in fall rates by medication class; thus, this analysis is outside the scope of the study.

Future research
The future research section could be expanded. The limitations already noted suggest multiple avenues for subsequent investigation, and offering concrete examples would strengthen this section.

RESPONSE: We have expanded the Future Research section and offer concrete examples (line 250): “In the context of a large-scale trial, process measures to elucidate clinician engagement with intervention materials could be collected and a range of medication outcomes (e.g., dose reduction, sustained discontinuation) could be assessed. Analyses of fall rates stratified by number and type of high-risk medication would also be informative.”

Engagement of persons with dementia and care partners
I appreciated the discussion around engaging both the person with dementia and their care partner, and encourage expansion of this point. The qualitative interviews provide compelling evidence; for example, one caregiver’s concern about sending extensive information directly to someone with dementia highlights the importance of involving care partners. In the conclusion, the manuscript states: “Care partner identification may not be necessary…however, engaging care partners may results in higher rates of deprescribing.” I recommend avoiding language that suggests care partners may be unnecessary. At least one caregiver explicitly questioned this approach, underscoring the value of incorporating care partners into the intervention.

RESPONSE: We have omitted the sentence in question from the Conclusions (line 255). We have expanded our Discussion around the issue of care partner identification through pragmatic means as follows (line 219): “Care partner information (e.g., name, contact information, relationship to the person with dementia) was not available in the health system at the time the present study was conducted. While we were able to deliver our intervention to those with dementia without identifying their care partner(s), our qualitative findings suggest that for some care partners, an intervention delivered to the person with dementia may raise concerns. Methods to identify care partners pragmatically from health system records are an area of active investigation. Care partner identification is an important consideration for future health-system-embedded pragmatic trials involving people with dementia.”